# Glycemic Dysregulation, Inflammation and Disease Outcomes in Patients Hospitalized with COVID-19: Beyond Diabetes and Obesity

**DOI:** 10.3390/v15071468

**Published:** 2023-06-28

**Authors:** Angelos Liontos, Dimitrios Biros, Aikaterini Kavakli, Rafail Matzaras, Ilias Tsiakas, Lazaros Athanasiou, Valentini Samanidou, Revekka Konstantopoulou, Ioannis Vagias, Aikaterini Panteli, Christiana Pappa, Nikolaos-Gavriel Kolios, Maria Nasiou, Eleni Pargana, Haralampos Milionis, Eirini Christaki

**Affiliations:** 11st Division of Internal Medicine & Infectious Diseases Unit, University General Hospital of Ioannina, Faculty of Medicine, University of Ioannina, 45500 Ioannina, Greece; angelosliontos@gmail.com (A.L.); dimitrisbiros@gmail.com (D.B.); rafail.matz@gmail.com (R.M.); il.tsiakas@gmail.com (I.T.); lazathanasiou@gmail.com (L.A.); valentinasmnsmn@gmail.com (V.S.); revekkakon@gmail.com (R.K.); doc_gvagias@yahoo.gr (I.V.); katerinapanteli@hotmail.com (A.P.); hmilioni@uoi.gr (H.M.); 2Faculty of Medicine, University of Ioannina, 45110 Ioannina, Greece; kavaklikaterina@gmail.com (A.K.); md06567@uoi.gr (C.P.); md06307@uoi.gr (N.-G.K.); md06549@uoi.gr (M.N.); md06655@uoi.gr (E.P.)

**Keywords:** SARS-CoV-2, COVID-19, diabetes mellitus, obesity, glycemic control, hyperglycemia

## Abstract

Introduction: During the COVID-19 pandemic, diabetes mellitus (DM) and obesity were associated with high rates of morbidity and mortality. The aim of this study was to investigate the relationship between markers of inflammation, disease severity, insulin resistance, hyperglycemia, and outcomes in COVID-19 patients with and without diabetes and obesity. Materials and Methods: Epidemiological, clinical, and laboratory data were collected from the University Hospital of Ioannina COVID-19 Registry and included hospitalized patients from March 2020 to December 2022. The study cohort was divided into three subgroups based on the presence of DM, obesity, or the absence of both. Results: In diabetic patients, elevated CRP, IL-6, TRG/HDL-C ratio, and TyG index, severe pneumonia, and hyperglycemia were associated with extended hospitalization. Increased IL-6, NLR, and decreased PFR were associated with a higher risk of death. In the obese subgroup, lower levels of PFR were associated with longer hospitalization and a higher risk of death, while severe lung disease and hyperglycemia were associated with extended hospitalization. In patients without DM or obesity severe pneumonia, NLR, CRP, IL-6, insulin resistance indices, and hyperglycemia during hospitalization were associated with longer hospitalization. Conclusion: Inflammatory markers and disease severity indices were strongly associated with disease outcomes and hyperglycemia across all subgroups.

## 1. Introduction

Severe acute respiratory syndrome coronavirus 2 (SARS-CoV-2), emerged in late 2019 causing novel coronavirus 2019 (COVID-19) disease, primarily affecting the respiratory system. As of the 20 April 2023, there have been 762 million confirmed cases and almost 7 million deaths reported globally due to COVID-19 [1]. Since the early stages of the pandemic, underlying comorbidities have been associated with a poorer prognosis of COVID-19 [2,3].

Among chronic cardiometabolic diseases, diabetes mellitus (DM), obesity, and cardiovascular disease were associated with higher morbidity and mortality rates in COVID-19 patients [4,5,6]. DM is a chronic condition, with multisystemic manifestations and a major health concern, affecting more than 8.8% of the adult population worldwide [7]. DM is one of the most common comorbidities in COVID-19 patients [8,9], linked to an increased risk of hospitalization and ICU admission (odds ratio [OR]: 2.79) [10]. It is estimated that the prevalence of DM increases with COVID-19 severity, ranging from 14.3% to 27% [2,11]. Furthermore, it was observed that COVID-19 diabetic patients exerted a higher mortality rate compared to non-diabetics (adjusted Hazard Ratio [aHR], 95% CI: 1.23 (1.14–1.32)) [12].

It is postulated that a cumulative effect of structural and functional molecular modifications may contribute to an increased susceptibility to COVID-19 [13]. Diabetic patients exhibit upregulation of angiotensin-converting enzyme-2 (ACE2), characterized by increased expression of this receptor in alveolar lung cells [13,14]. The ACE2 receptor acts as a cellular entry mediator for SARS-CoV-2, which may render diabetic patients more vulnerable to the virus [13,14]. Hyperglycemia can induce glycosylation of ACE2 receptors, thereby increasing viral entry, independent of receptor expression [15,16].

DM has also been associated with dysregulation of innate and adaptive immunity, caused by dysfunction of neutrophils, macrophages, and natural killer (NK) cells, as well as suppressed cytokine production [17]. Conversely, DM is linked to a chronic, pro-inflammatory state [15,18]. Therefore, individuals with DM are more susceptible to developing an exaggerated inflammatory response to SARS-CoV-2 infection [15,18]. It is worth noting that excessive production of circulating cytokines, coupled with pancreatic beta cell infection, may lead to insulin resistance and dysregulation of pre-existing DM, as well as the onset of new DM in individuals with prior normoglycemia [19].

Obesity can burden an individual’s overall health and has grown into an epidemic [20]. It is known to have a deleterious impact on respiratory function as it lowers ventilatory capacity and respiratory drive [21]. In addition, central obesity, as well as excess visceral fat, have a detrimental effect on the compliance of the chest wall and lungs [21]. Also, obesity-associated over-production of inflammatory cytokines impairs immune responses [21]. These reasons could explain the increased susceptibility of obese patients to respiratory infections (upper respiratory tract infections: adjusted OR 1.55 and lower respiratory tract infections: adjusted OR 2.02, respectively) [22]. Similarly, obesity has been linked with an increased risk of SARS-CoV-2 infection (OR: 2.42) and the disease’s progression to severe form (hospitalization, intubation, and death, OR:1.62, 1.75, and 1.23, respectively) [5,23]. The chronic inflammatory state [4] and immune dysfunction observed in individuals with obesity have been suggested as potential underlying mechanisms in the progression of COVID-19 [23]. Furthermore, adipocytes have been shown to exhibit higher ACE2 expression levels compared to the lung, potentially rendering them more vulnerable to viral entry [24,25]. However, the responsible underlying mechanisms are still unclear [5,6,26,27].

Glycemic dysregulation; namely hyperglycemia is common in COVID-19 patients (diabetics and non-diabetics) [11]. Hyperglycemia has been associated with disease severity and poorer outcomes in COVID-19 patients [5,6]. Growing evidence suggests that hyperglycemia serves as an independent risk factor for COVID-19 disease severity and outcomes regardless of prior history of DM or not [7]. Indeed, it has been shown that hyperglycemia in these patients increases the risk of mortality (7-fold) compared to normoglycemia [4,28]. This risk was even higher in non-diabetics compared to dia-betics with hyperglycemia [7]. It has also been shown that hyperglycemia is directly associated with a higher SARS-CoV2 replication rate in the host [13]. Thus, glucose monitoring and control may exert a key role as modifiable risk factor in COVID-19 [7]

The objective of this study was to examine the relationship between markers of inflammation, disease severity, insulin resistance, and outcomes in COVID-19 patients with and without DM and obesity. Additionally, we investigated the correlation between hyperglycemia and COVID-19-associated inflammation, severity, and outcomes in all patient groups.

## 2. Materials and Methods

### 2.1. Population

The population study included COVID-19 patients who were hospitalized at the University Hospital of Ioannina from March 2020 to December 2022. SARS-CoV-2 infection was diagnosed through RT-PCR. The patient cohort was divided into sub-groups, based on the presence of DM, obesity, or the absence of both.

### 2.2. Data Collection

This study was approved by the Institutional Ethics Committee of the University Hospital of Ioannina as a retrospective cohort study on hospitalized COVID-19 patients (Protocol Number: 5/11-03-2021 (issue: 3)/The University Hospital of Ioannina COVID-19 Registry, NCT05534074). Data on epidemiological, clinical, and laboratory parameters were obtained from the University Hospital of Ioannina COVID-19 Registry. As specified by the study protocol, personal identifier codes were used to import data anonymously into a digital database for each patient. Data collection adhered to European Guidelines for Good Clinical and Laboratory Practice and the Helsinki Declaration.

### 2.3. Study Variables and Endpoints

Study variables included baseline somatometric data and comorbidities; inflammatory markers (white blood cell (WBC) count, C-reactive protein (CRP), procalcitonin (PCT) and interleukin-6 (IL-6)); insulin resistance markers (triglycerides to HDL cholesterol (TRG/HDL-C) ratio and triglyceride–glucose (TyG) index); and disease severity markers; partial pressure of oxygen/fraction of inspired oxygen ratio/PO_2_/FiO_2_ ratio (PFR) and computed tomography (CT) burden of disease (CTBoD), defined as the percentage of the diseased lung parenchyma on CT of the lungs. The laboratory parameters examined in the study were documented on admission. An incidence of hyperglycemia was defined according to the ADA guidelines for hospitalized patients (≥140 mg/dL) and was documented on admission or during hospitalization (assessed at multiple time points; day: 3, 5, 7, 9, 11, and 15). Study outcomes included length of stay (LoS) > 7 days, intubation, and death during hospitalization. The association of hyperglycemia and disease inflammatory markers was also investigated.

### 2.4. Statistical Analysis

The preliminary analysis was performed with Mann–Whitney and chi-squared tests. Further, multivariate binary logistic regression adjusted for the patient’s age and sex was used for the analyses of outcomes. Of note, analyses concerning hyperglycemia were also adjusted for corticosteroid administration during hospitalization. All statistical analyses were performed using IBM SPSS Statistics 26.0 software.

## 3. Results

The studied population consisted of 1458 COVID-19 patients that were hospitalized at the University Hospital of Ioannina from March 2020 to December 2022. Table 1 summarizes the baseline demographic, somatometric characteristics, comorbidities, inflammatory markers, insulin resistance indices, disease severity markers, and outcomes. The multivariate binary logistic regression results for inflammatory markers, insulin resistance indices, disease severity markers, and outcomes for the diabetic, obese, and non-diabetic non-obese subgroups are presented in Table 2, Table 3 and Table 4, respectively. Table 5 displays the multivariate binary logistic regression results for inflammatory markers, insulin resistance indices, disease severity markers, and the incidence of hyperglycemia on admission and during hospitalization for each sub-group. Figure 1, Figure 2 and Figure 3 present distribution plots of inflammatory markers (NLR, PCT, IL-6), disease severity markers (PFR), and insulin resistance indices (TRG/HDL ratio, TyG index) for each sub-group of patients, respectively.

### 3.1. Patients with Diabetes

Demographic, somatometric baseline characteristics, comorbidities, inflammatory markers, insulin resistance indices, disease severity markers, and outcomes are summarized in Table 1. A total of 270 patients were diabetics and 165 of them were male. Patients’ mean age and BMI were 73.45 years and 25.54 kg/m^2^, respectively. The most frequent comorbidities in this subpopulation were arterial hypertension, dyslipidemia, and coronary artery disease. In the diabetic population, increased levels of CRP, IL-6, and CT BoD, as well as lower levels of PFR, were associated with a greater length of hospital stay (LoS). Also, hyperglycemia on admission and hospital-acquired hyperglycemia, as well as higher levels of TRG/HDL-C ratio and TyG index were associated with more extended hospitalization. Higher levels of CT BoD and lower levels of PFR were associated with a significantly higher incidence of intubation. Finally, elevated levels of IL-6, increased neutrophil to lymphocyte ratio (NLR), and decreased PFR were associated with a higher risk for death (Table 2 and Table 5).

### 3.2. Patients with Obesity

Table 1 summarizes baseline characteristics, comorbidities, inflammatory and insulin resistance markers, disease severity, and outcomes. The obese population consisted of 201 patients, 107 of which were male. Their mean age and BMI were 59.69 years and 34.24 kg/m^2^. The most frequent comorbidities in this subgroup were arterial hypertension, dyslipidemia, and thyroid disturbances. Lower levels of PFR were highly associated with longer hospitalization and higher rates of death, while increased CT BoD, NLR, and hyperglycemia during hospitalization were associated with a greater LoS. Increased levels of the TyG index were also associated with a higher risk for hyperglycemia on admission and during hospitalization (Table 3 and Table 5).

### 3.3. Patients without Diabetes or Obesity

Demographic, somatometric baseline characteristics, comorbidities, inflammatory markers, insulin resistance indices, disease severity markers, and outcomes are summarized in Table 1. There was a total of 631 COVID-19 patients without diabetes or obesity, 262 of which were female. The most frequent comorbidities were arterial hypertension, dyslipidemia, and coronary artery disease. Increased CT BoD, NLR, CRP, IL-6, PCT, insulin resistance indices, and hyperglycemia during hospitalization, as well as decreased PFR were associated with longer hospitalization. Also, hyperglycemia during hospitalization and increased levels of insulin resistance indices were associated with longer hospital stays. Similar results were noted for intubation and death. In addition to lower PFR, elevated levels of NLR, CRP, IL-6, and TyG index were highly associated with an increased incidence of hyperglycemia on admission. Similar associations were documented for increased levels of CRP and TyG index and hyperglycemia during hospitalization (Table 4 and Table 5).

## 4. Discussion

In this study, we assessed the association between inflammatory and disease severity markers, as well as insulin resistance indices and outcomes in patients with metabolic disorders, specifically, diabetes and obesity. We also investigated these associations in patients without diabetes or obesity. Additionally, we explored the association between inflammatory and disease severity markers, insulin resistance indices, and hyperglycemia on admission or during hospitalization for each patient sub-group.

Diabetes, which ranks seventh among the leading causes of mortality worldwide, is associated with vascular complications that have a significant impact on quality of life [29]. The clinical understanding of the link between diabetes and susceptibility to infection has been well-established for a long time [30]. Diabetes is a chronic inflammatory condition that affects both metabolic and vascular functions, and is therefore likely to contribute to an increased susceptibility to infection [31]. Elderly patients with type 2 diabetes are at an increased risk of severe complications from infections, including pneumonia and influenza [32]. Our study demonstrated that COVID-19 patients with diabetes had longer hospital stays when they exhibited elevated levels of CRP, IL-6, and CT BoD, as well as decreased PFR. Greater lung damage, as indicated by higher CT BoD and lower PFR, also increased the risk of intubation. Among diabetic patients, a higher risk of death was associated with elevated IL-6 and NLR levels, as well as decreased PFR.

There is mounting evidence indicating that SARS-CoV-2 worsens inflammation in patients with diabetes [31]. Diabetes is characterized by a chronic inflammatory state, leading to dysfunction of immune cells, such as inhibition of phagocytosis and neutrophil chemotaxis, which impairs the ability to counteract infectious organisms [33,34]. Diabetic patients frequently demonstrate compromised T cell-mediated immunity and delayed hypersensitivity [34]. Studies on COVID-19 patients suggest alterations in CD4 and CD8 cell counts, indicating apoptosis and lymphocytopenia [31]. Additionally, a recent analysis has shown that diabetics exhibit higher levels of ACE2 expression in lung cells, thus providing support for the hypothesis that SARS-CoV-2 has a higher affinity for infecting lung cells in these individuals [35]. This upregulation of ACE2 can increase viral binding and facilitate viral entry into host cells, resulting in severe lung damage [35]. Furthermore, an elevation in levels of proinflammatory cytokines such as IL-6, serum ferritin, and CRP was observed, with IL-6 serving as a reliable predictor of COVID-19 severity and progression [36,37,38]. The overproduction of cytokines, i.e., the cytokine storm, plays a crucial role in the progression of SARS-CoV-2 infection, leading to hyperinflammation and dysfunction of end-organs [39,40].

Notably, our study found that glycemic dysregulation either upon admission or during hospitalization, as well as elevated levels of TRG/HDL-C ratio and TyG index, were linked with prolonged hospital stay in the patient cohort. The latter ratios have been adopted by researchers as indices of insulin resistance resulting in imbalances of glucose metabolism [41,42,43]. Also, TyG index and TRG/HDL-C have been proposed as predictors of glycemic control in normoweight, overweight and obese patients with DM [44]. Similar to our findings, the positive association of these indices was also observed in a study (*n* = 65) of elderly diabetic population. In this cohort of patients elevated glucose levels were associated with a high TyG index andTRG/HDL-C ratio compared with glucose levels prior to SARS-CoV-2 infection [45]. TyG index and TRG/HDL-C ratio have been also proposed as prognostic factors of disease severity in COVID-19 patients. In a study of 1228 patients (diabetics and non-diabetics), it was shown that patients with severe disease exhibited significantly elevated levels of TyG index and TRG/HDL-C ratio (*p* < 0.05). On the other hand, survivors exhibited significantly lower TRG/HDL-C ratio and TyG index levels compared to non-survivors (*p* < 0.05). Investigators also found that the TyG index and TRG/HDL-C ratio were significant predictors of severity (OR = 1.42 and OR = 1.06, respectively). Similarly, these indices were associated with COVID-19 mortality (TRG/HDL-C ratio: OR = 1.12, TyG index: OR = 1.52, respectively) [43].

Hyperglycemia is a significant predictor of increased severity and mortality in individuals infected with H1N1, SARS-CoV, and MERS-CoV. However, studies showed conflicting results regarding the impact of hyperglycemia in individuals infected with SARS-CoV-2 [46,47,48,49]. SARS-CoV-2 infection may complicate diabetes by increasing glucose levels, resulting in oxidative stress and inflammation. Viral binding to ACE2 on acinar cells results in tissue damage and inhibition of lymphocyte proliferation due to hyperglycemia [31]. SARS-CoV-2 infection in diabetics induces elevated stress and glucose levels through the release of glucocorticoids and catecholamines [50]. The latter have been found to hinder immune function, thus heighten susceptibility to infections [31]. As observed in other viral infections, hyperglycemia facilitates viral replication and augment viral load resulting in tissue damage [51,52,53]. Moreover, hyperglycemia has been found to suppress antiviral immune response, as demonstrated in previous studies of influenza infection. This suppression can lead to pulmonary dysfunction and fatal outcomes in DM [33]. In addition, delayed clearance of SARS-CoV-2 observed in these patients may exacerbate the severity and progression of COVID-19. Overall, studies indicate that patients with diabetes experience a more severe form of COVID-19 owing to heightened inflammation. This is due to several factors, including increased levels of ACE2, reduced T cell function or lymphocytopenia, hyperinflammation or cytokine storm, impaired monocyte/macrophage function, increased coagulation, and delayed viral clearance [31].

Of note, it has been shown that hyperglycemia on admission is associated with worse outcomes in COVID-19. In a study of 11,312 patients, elevated glucose were associated with significantly higher mortality rates compared to normoglycemia (*p* < 0.001), regardless of pre-existing diabetes. Hyperglycemia was found to be an independent risk factor of mortality (glucose levels > 180 mg/dL: HR 1.50, 140–180 mg/dLQ HR 1.48). Hyperglycemia was also associated with increased risk of intubation and intensive care unit (ICU) admission and mortality (glucose levels > 180 mg/dL: OR 2.02, *p* < 0.001, 140–180 mg/dL: OR 1.70, *p* < 0.001 compared to glucose levels < 140 mg/dL) [54]. The positive association of admission hyperglycemia and worse outcomes in COVID-19 was confirmed by other studies. In the study by Wang et al., (*n* = 695) elevated glucose levels > 126 mg/dL on admission was an independent risk factor for 28-day mortality in patients without known DM [55]. Furthermore, in the study by Wu et al., (*n* = 2041), elevated glucose levels (≥110 mg/dL) on admission were an independent risk factor for critical disease (ICU admission, mechanical ventilation, hemodynamic instability) or death (OR 1.30, *p* = 0.026). Higher glucose levels in critical patients were also associated with higher in-hospital mortality (OR 2.39, *p* = 0.001) [56]. Similar to these, in our study admission hyperglycemia (glucose levels > 140 mg/dL) was associated with worse outcomes of COVID-19 disease. Across all three sub-groups of patients, hyperglycemia on admission was associated with longer hospital stay. Hyperglycemia has been also associated with higher levels of inflammatory markers. In the study by Geetha et al., (*n* = 520) inflammatory markers (i.e., CRP, ferritin, d-dimers) were higher in patients with hyperglycemia compared to those with normal glucose levels [57]. Similarly, mortality rate and length of stay was higher in the hyperglycemic group [57]. Our study shares similar results. Elevated inflammatory markers were associated with a higher risk of hyperglycemia and concomitant worse prognosis.

Notably, SARS-CoV-2 infection can cause temporary hyperglycemia in healthy individuals, similar to what was observed in SARS-CoV-1 infection [37,58]. Specifically, hyperglycemia and new-onset DM have been previously reported in hospitalized patients with COVID-19 [11]. Similarly, during the COVID-19 pandemic, it has been suggested that the patients without a pre-existing history of DM have an increased risk of developing the disease [59].

Stress hyperglycemia has been linked to relative insulin deficiency and elevated levels of circulating free fatty acids, which are commonly observed in acute illnesses sush as severe infections or myocardial infarction [11,60]. Stress-induced hyperglycemia appears to have a more negative impact on the prognosis of non-diabetic COVID-19 hospitalized patients compared to diabetics. Also, stress-induced hyperglycemia is a more reliable prognostic marker for COVID-19 patients than their glycemic status prior to hospitalization [61,62]. SARS-CoV-2 induced cytokine storm is a highly inflammatory and prothrombotic condition that can affect pancreatic β-cells both directly and indirectly [11]. Infection of β-cells in COVID-19 patients results in reduced insulin levels and secretion, as well as cell apoptosis. Viral entry results to pancreatic endocrine dysfunction characterized by reduced insulin secretion in response to glucose and triggers the self-destruction mechanism of insulin-producing cells (through kinase pathways) [9,11]. Thus, the cytokine storm in COVID-19 may exacerbate stress hyperglycemia as acute inflammation may further worsen insulin resistance [11,63]. Among these pathways, it has also been proposed that SARS-CoV-2 infection may initiate an autoimmune response targeting β-cells [9,64,65]. Last, microvascular dysfunction is a probable cause of hyperglycemia in COVID-19 patients as it may hinder glucose disposal. This relationship between microvascular function and glycemic control is reciprocal and well established [66].

Obesity is an identified risk factor for both susceptibility and severity of COVID-19 disease. Studies have demonstrated that severe obesity (BMI  >  40 kg/m^2^) is linked to higher rates of hospitalization, critical care admission, and fatalities during past viral pandemics [21]. Obesity serves as an independent risk factor for severe COVID-19, and recent research indicates that visceral obesity may also increase the likelihood of complications [67]. Recent analysis showed that correlation between COVID-19 mortality and obesity burden in a country’s adult population remains strong [6]. In the group of patients with obesity, respiratory failure as evidenced by lower PFR were strongly associated with longer hospitalization and higher rates of death, while increased CT BoD, NLR, and hyperglycemia during hospitalization were associated with longer hospitalization. Obesity is an established risk factor for DM that is linked to insulin resistance [68]. Obese individuals release elevated levels of non-esterified fatty acids, glycerol, hormones, and pro-inflammatory cytokines from adipose tissue, which may contribute to the onset of insulin resistance [68]. As it has already been demonstrated [41,42,43,44,45], increased TyG index was also associated with a higher risk for hyperglycemia on admission and during hospitalization. This is in accordance with a study by Chen et al., where COVID-19 patients without pre-existent DM, presented insulin resistance upon hospital admission t, as indicated by the increased TyG index levels [69], demonstrating that SARS-CoV-2 infection predisposes to insulin resistance even in non-diabetic patients [69].

One interesting correlation stemming from our study and requiring further exploration, is between NLR and hyperglycemia. NLR was associated with hyperglycemia on admission in all three sub-groups in our cohort. NLR is a novel inflammatory marker indicative of a significant inflammatory burden in specific diseases [70]. It has been shown that elevated NLR levels may suggest impaired glucose metabolism and can serve as an additional marker of glycemic control level, in individuals with type 2 DM [70]. In a study of 271 patients, participants were allocated to three groups based on admission glycemic status. In the first group patients had glucose levels < 140 mg/dL (normoglycemic, NG) and the second group included patients with known DM [71]. Patients in the third group (hyperglycemic, HG) had no prior history of DM and glucose levels ≥ 140 mg/dL. Data analysis showed that neutrophil count was higher, whereas lymphocyte count and PaO2/FiO2 were lower in HG than in DM and NG patients. DM and HG patients had higher D-dimer and worse inflammatory profile. Mortality was greater in HG (39.4% vs. 16.8%; unadjusted hazard ratio [HR] 2.20, 95% CI 1.27–3.81, *p* = 0.005) than in NG (16.8%) and marginally so in DM (28.6%; 1.73, 0.92–3.25, *p* = 0.086) patients. Upon multiple adjustments, only HG remained an independent predictor (HR 1.80, 95% CI 1.03–3.15, *p* = 0.04) [71].

As it has already been demonstrated during the pandemic, in large population studies, severe lung damage and respiratory failure, as well as excessive inflammation, is linked to longer hospital stay and worse outcomes [38,72,73,74,75,76,77], which is in concordance with our results. Our study revealed that glycemic dysregulation on admission or during hospitalization, as well as higher levels of TRG/HDL-C ratio and TyG index, were associated with prolonged hospital stay. Notably, insulin resistance indices and hyperglycemia during hospitalization were linked to extended hospitalization in non-diabetic non-obese patients as well. Hyperglycemia on admission was correlated with a lower PFR, elevated levels of NLR, CRP, IL-6, and TyG index, while hyperglycemia during hospitalization was associated with increased levels of CRP and TyG index. It is worth noting that corticosteroid use did not affect this association, as all hospitalized patients requiring supplementary oxygen were receiving corticosteroids, and this variable was included in the regression analysis.

However, the precise impact of the virus on insulin secretion and glycemic control, as well as the correlation between glucose variability and SARS-CoV-2 virulence, are still uncertain.

## 5. Conclusions

In our study, elevated levels of inflammatory markers, respiratory failure, and COVID-19-associated lung injury were associated with a higher incidence of hyperglycemia in patients with diabetes and obesity, but also those without either disease. Similarly, across all groups, hyperglycemia was associated with longer hospital stays, and glycemic dysregulation increased the risk of worse outcomes. Finally, it was observed that insulin resistance assessed by novel markers serves as a strong predictor of both worse regulation of glucose levels and disease outcomes in COVID-19 patients.

## Figures and Tables

**Figure 1 viruses-15-01468-f001:**
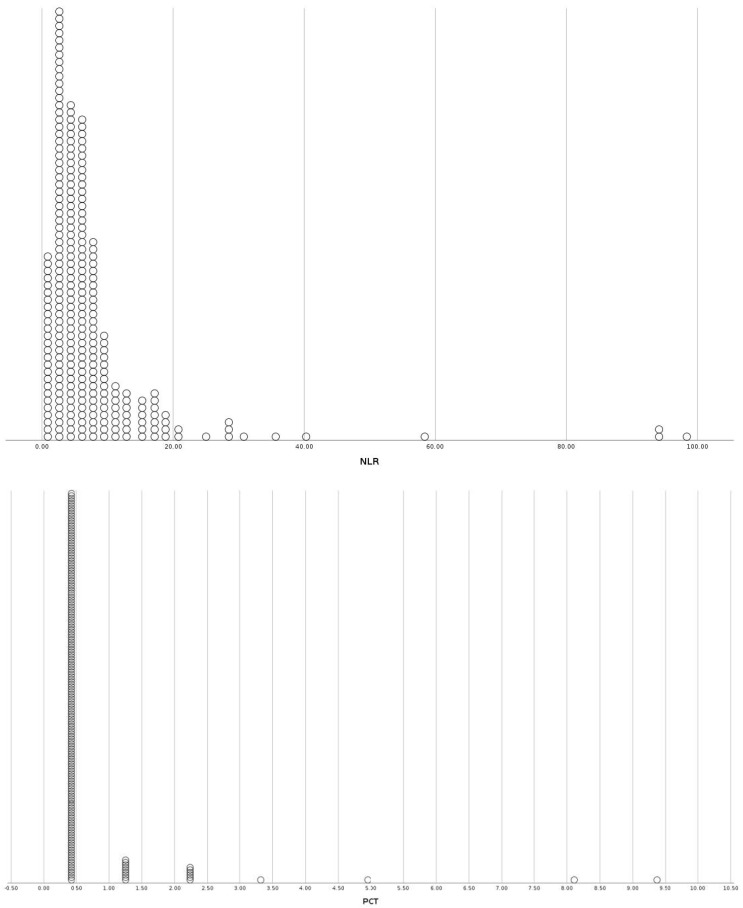
Distribution plots of inflammatory markers (NLR, PCT, IL-6), disease severity markers (PFR) and insulin resistance indices (TRG/HDL ratio, TyG index) for the sub-group of patients with diabetes. NLR: Neutrophil to Lymphocyte Ratio, PCT: Procalcitonin, IL-6: Interleukin-6, PFR: PO2/FiO2 Ratio, TRG/HDL-C: Triglycerides to HDL-Cholesterol ratio, TyG: Triglyceride-glucose index.

**Figure 2 viruses-15-01468-f002:**
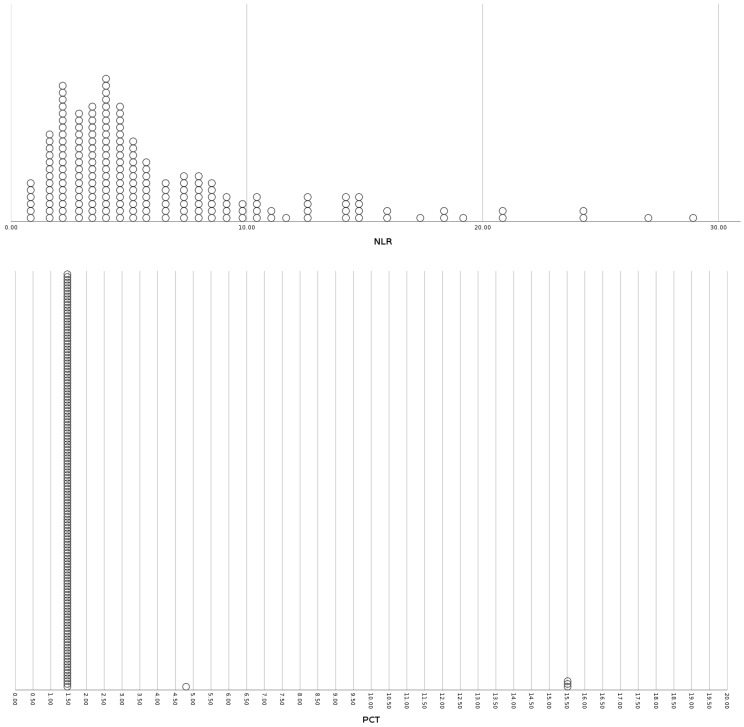
Distribution plots of inflammatory markers (NLR, PCT, IL-6), disease severity markers (PFR) and insulin resistance indices (TRG/HDL ratio, TyG index) for the sub-group of patients with obesity. NLR: Neutrophil to Lymphocyte Ratio, PCT: Procalcitonin, IL-6: Interleukin-6, PFR: PO2/FiO2 Ratio, TRG/HDL-C: Triglycerides to HDL-Cholesterol ratio, TyG: Triglyceride-glucose index.

**Figure 3 viruses-15-01468-f003:**
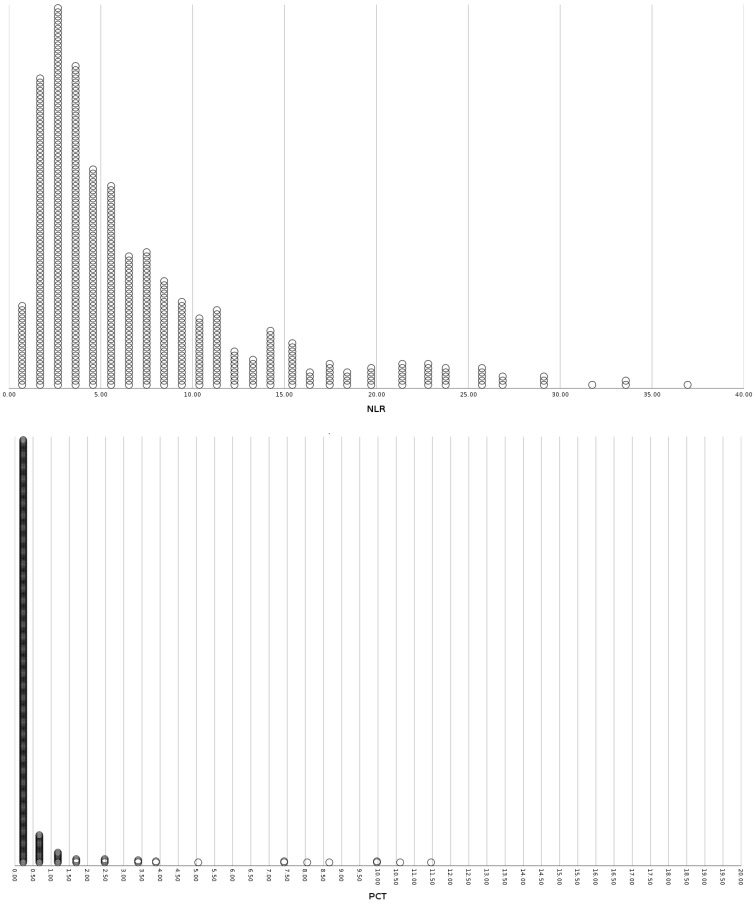
Distribution plots of inflammatory markers (NLR, PCT, IL-6), disease severity markers (PFR) and insulin resistance indices (TRG/HDL ratio, TyG index) for the sub-group of patients without diabetes or obesity. NLR: Neutrophil to Lymphocyte Ratio, PCT: Procalcitonin, IL-6: Interleukin-6, PFR: PO2/FiO2 Ratio, TRG/HDL-C: Triglycerides to HDL-Cholesterol ratio, TyG: Triglyceride-glucose index.

**Table 1 viruses-15-01468-t001:** Demographic, somatometric baseline characteristics, comorbidities, inflammatory markers, insulin resistance indices, disease severity markers, and outcomes for each studied sub-group. Data presented as frequency or mean/standard deviation.

Cohort	Patients with Diabetes		Patients with Obesity		Patients without Diabetes or Obesity	
Characteristics	Frequency (*n*)	Mean/SD	Frequency (*n*)	Mean/SD	Frequency (*n*)	Mean/SD
Sex (male/female)	165/105		107/94		369/262	
Age (years)		73.45/12.31		59.69/16.33		63.85/18.34
BMI (kg/m^2^)		25.54/2.93		34.24/4.35		25.64/2.79
Plasma glucose (mg/dL)		177.92/82.37		126.22/45.06		119.73/35.08
WBC count (#/μL)		8205.65/7312.56		7218.68/3762.28		7244.40/4032.51
NLR		7.91/11.50		6.19/5.08		7.41/7.93
CRP (mg/L)		118.05/121.33		124.11/117.38		112.02/115.81
PCT (ng/mL)		0.73/3.08		1.36/11.04		0.53/2.24
IL-6 (pg/mL)		61.32/156.61		34.52/39.38		56.90/174.61
PFR		264.73/123.21		260.56/109.97		281.78/123.93
TRG/HDL-C ratio		4.66/8.85		3.51/2.92		3.20/2.18
TyG index		9.11/0.72		8.75/0.59		8.66/0.55
Hyperglycemia on admission	165		46		118	
Hyperglycemia during hospitalization	229		100		317	
Comorbidities						
CAD	80		21		118	
AH	192		86		248	
Dyslipidemia	153		51		191	
COPD	29		15		45	
Thyroid disease	31		27		64	
Autoimmune disease	15		14		43	
Cancer	30		9		58	
Smoking (active)	36		25		88	
COVID-19 vaccination	124		67		186	
Outcomes						
LoS > 7 days	145		111		320	
Intubation	14		5		31	
Patient death	39		12		68	

BMI: body mass index; WBC count: white blood cell count; NLR: neutrophil to lymphocyte ratio; CRP: C-reactive protein; PCT: procalcitonin; IL-6: interleukin-6; PFR: PO_2_/FiO_2_ ratio; TRG/HDL-C: triglycerides to HDL-cholesterol ratio; TyG: triglyceride–glucose index; CAD: coronary artery disease; AH: arterial hypertension; COPD: chronic obstructive pulmonary disease; LoS: length of stay.

**Table 2 viruses-15-01468-t002:** Multivariate binary logistic regression results of inflammatory markers, insulin resistance indices, disease severity markers, and outcomes for the sub-group of patients with diabetes.

Outcomes	LoS > 7 Days		Intubation		Death	
Variables	OR	*p*-Value	OR	*p*-Value	OR	*p*-Value
WBC count (#/μL)	1.00	0.407	1.000	0.959	1.000	0.004
NLR	1.01	0.235	1.002	0.951	1.035	0.005
CRP (mg/L)	1.01	0.186	1.003	0.028	1.004	0.001
PCT (ng/mL)	0.985	0.763	0.967	0.846	0.956	0.663
IL-6 (pg/mL)	1.004	0.074	1.002	0.038	1.002	0.078
PFR	0.996	0.001	0.990	0.005	0.994	0.002
CT BoD	4.980	<0.001	17.634	0.009	2.753	0.057
NLR > 3.1	1.099	0.737	2.36	0.271	2.932	0.035
CRP > 100	1.787	0.022	3.321	0.051	1.913	0.073
PCT > 0.5	1.554	0.278	2.654	0.198	1.876	0.189
IL-6 > 24	2.840	0.001	2.925	0.129	5.582	<0.001
PFR < 200	4.413	<0.001	6.636	0.007	5.984	<0.001
Hyperglycemia on admission	1.719	0.034	1.964	0.389	1.896	0.111
Hyperglycemia during hospitalization	3.704	0.001	0.995	0.995	1.097	0.861
TRG/HDL-C ratio	1.209	0.005	0.992	0.890	0.999	0.973
TyG index	2.065	0.003	1.492	0.383	1.784	0.039

WBC count: white blood cell count; NLR: neutrophil to lymphocyte ratio; CRP: C-reactive protein; PCT: procalcitonin; IL-6: interleukin-6; PFR: PO_2_/FiO_2_ ratio; TRG/HDL-C: triglycerides to HDL-cholesterol ratio; TyG: triglyceride–glucose index; CT BoD: CT burden of disease; LoS: length of stay; OR: odds ratio.

**Table 3 viruses-15-01468-t003:** Multivariate binary logistic regression results of inflammatory markers, disease severity markers, and the study outcomes for the sub-group of patients with obesity.

Outcomes	LoS > 7 Days		Intubation		Death	
Variables	OR	*p*-Value	OR	*p*-Value	OR	*p*-Value
WBC count (#/μL)	1.000	0.189	1.000	0.221	1.000	0.144
NLR	1.055	0.087	0.947	0.630	1.051	0.373
CRP (mg/L)	1.003	0.021	1.00	0.927	1.000	0.956
PCT (ng/mL)	0.929	0.354	0.199	0.700	1.041	0.112
IL-6 (pg/mL)	1.012	0.032	1.000	0.984	1.006	0.451
PFR	0.995	0.001	0.980	0.023	0.991	0.025
CT BoD	9.938	<0.001	*	*	1.311	0.835
NLR > 3.1	2.179	0.019	1.312	0.812	4.933	0.148
CRP > 100	1.624	0.096	1.521	0.653	1.314	0.674
PCT > 0.5	0.731	0.572	*	*	1.658	0.605
IL-6 > 24	1.831	0.063	3.226	0.317	2.383	0.248
PFR < 200	3.073	0.001	*	*	4.750	0.025
Hyperglycemia on admission	1.442	0.333	*	*	0.509	0.433
Hyperglycemia during hospitalization	2.316	0.009	*	*	0.587	0.467
TRG/HDL-C ratio	0.948	0.376	0.812	0.582	1.134	0.125
TyG index	1.292	0.374	0.542	0.432	0.818	0.756

WBC count: white blood cell count; NLR: neutrophil to lymphocyte ratio; CRP: C-reactive protein; PCT: procalcitonin; IL-6: interleukin-6; PFR: PO_2_/FiO_2_ ratio; TRG/HDL-C: triglycerides to HDL-cholesterol ratio; TyG: triglyceride–glucose index; CT BoD: CT burden of disease; LoS: length of stay; OR: odds ratio,* unable to calculate the odds ratios due to small sample size.

**Table 4 viruses-15-01468-t004:** Multivariate binary logistic regression results of inflammatory markers, disease severity markers, and the study outcomes for the sub-group of patients without diabetes or obesity.

Outcomes	LoS > 7 Days		Intubation		Death	
Variables	OR	*p*-Value	OR	*p*-Value	OR	*p*-Value
WBC count (#/μL)	1.000	0.346	1.000	0.259	1.000	0.002
NLR	1.024	0.034	1.038	0.024	1.062	<0.001
CRP (mg/L)	1.003	0.001	1.004	0.002	1.005	<0.001
PCT (ng/mL)	1.061	0.272	1.047	0.448	1.156	0.008
IL-6 (pg/mL)	1.000	0.858	1.001	0.087	1.003	<0.001
PFR	0.997	<0.001	0.989	<0.001	0.994	<0.001
CT BoD	4.757	<0.001	28.430	<0.001	5.535	<0.001
NLR > 3.1	1.511	0.022	5.552	0.021	5.724	<0.001
CRP > 100	2.179	<0.001	2.579	0.015	3.022	<0.001
PCT > 0.5	1.393	0.279	2.263	0.109	4.742	<0.001
IL-6 > 24	2.323	<0.001	2.496	0.041	3.705	<0.001
PFR < 200	2.442	<0.001	5.100	<0.001	3.546	<0.001
Hyperglycemia on admission	1.023	0.916	2.390	0.029	1.667	0.103
Hyperglycemia during hospitalization	1.685	0.002	3.352	0.010	1.425	0.226
TRG/HDL-C ratio	1.249	<0.001	1.186	0.023	1.369	<0.001
TyG index	2.418	<0.001	2.983	0.002	3.203	<0.001

WBC count: white blood cell count; NLR: neutrophil to lymphocyte ratio; CRP: C-reactive protein; PCT: procalcitonin; IL-6: interleukin-6; PFR: PO_2_/FiO_2_ ratio; TRG/HDL-C: triglycerides to HDL-cholesterol ratio; TyG: triglyceride–glucose index; CT BoD: CT burden of disease; LoS: length of stay; OR: odds ratio.

**Table 5 viruses-15-01468-t005:** Multivariate binary logistic regression results of inflammatory markers, disease severity markers, and the incidence of hyperglycemia on admission and during hospitalization for each sub-group.

Group of Patients	Patients with Diabetes				Patients with Obesity				Patients without Diabetes or Obesity			
Glycemic Status	Hyperglycemia on Admission		Hyperglycemia during Hospitalization		Hyperglycemia on Admission		Hyperglycemia during Hospitalization		Hyperglycemia on Admission		Hyperglycemia during Hospitalization	
Variables	OR	*p*-Value	OR	*p*-Value	OR	*p*-Value	OR	*p*-Value	OR	*p*-Value	OR	*p*-Value
WBC count (#/μL)	1.000	0.161	1.000	0.916	1.000	0.068	1.000	0.995	1.000	< 0.001	1.000	0.018
NLR	1.105	0.001	1.079	0.083	1.068	0.038	1.041	0.193	1.066	< 0.001	1.021	0.075
CRP (mg/L)	1.002	0.105	1.000	0.753	1.000	0.980	1.002	0.236	1.002	0.004	1.002	0.042
PCT (ng/mL)	1.256	0.236	3.472	0.186	0.797	0.541	0.507	0.175	1.004	0.934	1.012	0.792
IL-6 (pg/mL)	1.001	0.561	1.003	0.388	1.002	0.690	0.999	0.740	1.000	0.381	1.000	0.718
PFR	0.998	0.110	0.997	0.065	0.998	0.200	1.000	0.799	0.997	0.008	1.000	0.595
CT BoD > 50%	1.220	0.608	7.482	0.059	1.643	0.254	1.221	0.604	1.586	0.119	1.316	0.255
NLR > 3.1	0.997	0.992	0.755	0.499	1.885	0.172	1.260	0.512	2.402	0.002	1.369	0.105
CRP > 100 mg/L	1.638	0.060	1.793	0.116	0.716	0.350	1.321	0.357	1.882	0.003	1.441	0.035
PCT > 0.5 ng/mL	2.162	0.080	2.786	0.186	0.711	0.630	0.206	0.015	1.516	0.215	0.997	0.993
IL-6 > 24 pg/mL	0.919	0.782	1.239	0.634	0.745	0.453	0.900	0.752	0.583	0.027	0.807	0.283
PFR < 200	1.633	0.105	3.285	0.022	1.124	0.760	1.000	1.000	2.131	0.001	1.352	0.139
TRG/HDL-C ratio	0.992	0.649	1.313	0.203	1.048	0.441	0.968	0.582	1.007	0.890	1.064	0.200
TyG index	13.838	<0.001	4.005	0.001	34.014	<0.001	4.190	<0.001	9.408	<0.001	3.612	<0.001

WBC count: white blood cell count; NLR: neutrophil to lymphocyte ratio; CRP: C-reactive protein; PCT: procalcitonin; IL-6: interleukin-6; PFR: PO_2_/FiO_2_ ratio; TRG/HDL-C: triglycerides to HDL-cholesterol ratio; TyG: triglyceride–glucose index; CT BoD: CT burden of disease; OR: odds ratio.

## Data Availability

Data sharing is not applicable to this article.

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
