# Peer review of "Glycemic Dysregulation, Inflammation and Disease Outcomes in Patients Hospitalized with COVID-19: Beyond Diabetes and Obesity"

_viruses, 2023, doi:10.3390/v15071468_

Round 1

Reviewer 1 Report

This is an interesting study examining the effects of DM, obesity, and hyperglycemia on outcomes in SARS-CoV-2 infections. Similarly to other efforts HG, DM, and obesity result in increased levels of inflammatory markers, which exacerbates control and clearance of the viral infection.

A few comments:

(1) How relevant do the authors think upregulation of ACE2 in DM and HG is to outcome? There are many other reasons for worse outcomes, many of which the authors note, and ACE2 level increase may be somewhat simplistic.

(2) Can the authors comment more on the observation of transient HG as a consequence of SARS-CoV-2 infection, as it is in some patients with SARS-CoV? This was briefly mentioned at the end of the Discussion, and some more detailed comments with strengthen the manuscript.

Author Response

REVIEWER 1

Comments and Suggestions for Authors

This is an interesting study examining the effects of DM, obesity, and hyperglycemia on outcomes in SARS-CoV-2 infections. Similarly, to other efforts HG, DM, and obesity result in increased levels of inflammatory markers, which exacerbates control and clearance of the viral infection.

Comment #1

How relevant do the authors think upregulation of ACE2 in DM and HG is to outcome? There are many other reasons for worse outcomes, many of which the authors note, and ACE2 level increase may be somewhat simplistic.

Answer to Comment #1

We would like to thank the reviewer for his/her comment. As noted in our manuscript worse outcomes in DM and HG are multifactorial including a chronic inflammatory state in diabetics, cell mediated immunity dysfunction, ACE2 upregulation and overproduction of inflammatory cytokines.

In the discussion section, we first addressed the chronic inflammatory state in diabetics, which leads to both metabolic and vascular dysfunction, and may predispose to an increased susceptibility to infections.  Of note, elderly diabetic patients are at an increased risk of severe complications from infections (i.e. pneumonia and influenza) (page 13). Similarly, SARS-CoV-2 worsens inflammation in people with diabetes (page 13). Possible underlying mechanisms for the latter in diabetics include dysfunction of cell immunity such as inhibition of phagocytosis and neutrophil chemotaxis, which impairs the ability to eradicate infectious organisms and impairment of T cell-mediated immunity. In addition, it has been shown that COVID-19 results in apoptosis and lymphocytopenia, reflected in decreases in CD4 and CD8 cell counts (page 13).

Regarding ACE2 expression in lung cells, it has been shown that diabetics exhibit higher levels, supporting the hypothesis that SARS-CoV-2 has a higher affinity for infecting lung cells in these individuals. This upregulation of ACE2 increases viral binding and facilitate viral entry into host cells, resulting in severe lung damage in these patients (page 13).

The elevation of proinflammatory cytokines (i.e. IL-6, serum ferritin, and CRP) serves as a reliable predictor of COVID-19 severity and progression. Overproduction of cytokines (cytokine storm), seems to play a crucial role in the progression of SARS-CoV-2 infection, leading to hyperinflammation and end-organ dysfunction (page 13).

Comment #2

Can the authors comment more on the observation of transient HG as a consequence of SARS-CoV-2 infection, as it is in some patients with SARS-CoV? This was briefly mentioned at the end of the Discussion, and some more detailed comments with strengthen the manuscript.

Answer to Comment #2

We would like to thank the reviewer for his/her comment. We have added a more detailed comment in the discussion section (pages 14-15), on the observation of transient HG as a consequence of SARS-CoV-2 infection, also including relevant references. More specifically we have added the following paragraph:

Notably, SARS-CoV-2 infection can cause temporary hyperglycemia in healthy individuals, similarly to what was observed in SARS-CoV-1 infection (Liu, F., et al., ACE2 Expression in Pancreas May Cause Pancreatic Damage After SARS-CoV-2 Infection. Clin Gastroenterol Hepatol, 2020. 18(9): p. 2128-2130.e2., Yang, X., et al., Clinical course and outcomes of critically ill patients with SARS-CoV-2 pneumonia in Wuhan, China: a single-centered, retrospective, observational study. Lancet Respir Med, 2020. 8(5): p. 475-481). Specifically, hyperglycemia and new-onset DM have been previously reported in hospitalized patients with COVID-19 (Khunti, K., et al., COVID-19, Hyperglycemia, and New-Onset Diabetes. Diabetes Care, 2021. 44(12): p. 2645-2655). Similarly, during the COVID-19 pandemic, it has been suggested that patients without pre-existing history of DM have an increased risk of developing the disease (Al-Aly, Z. and Y. Xie, High-dimensional characterization of post-acute sequelae of COVID-19. 2021. 594(7862): p. 259-264).

Stress hyperglycemia has been linked to relative insulin deficiency and elevated levels of circulating free fatty acids, which are commonly observed in acute illnesses such as severe infections or myocardial infarction (Khunti, K., et al., COVID-19, Hyperglycemia, and New-Onset Diabetes. Diabetes Care, 2021. 44(12): p. 2645-2655, Capes, S.E., et al., Stress hyperglycaemia and increased risk of death after myocardial infarction in patients with and without diabetes: a systematic overview. Lancet, 2000. 355(9206): p. 773-8). Stress-induced hyperglycemia appears to have a more negative impact on the prognosis of non-diabetic COVID-19 hospitalized patients compared to diabetics. Also, stress-induced hyperglycemia is a more reliable prognostic marker for COVID-19 patients than their glycemic status prior to hospitalization (Mondal, S., et al., Stress hyperglycemia ratio, rather than admission blood glucose, predicts in-hospital mortality and adverse outcomes in moderate-to severe COVID-19 patients, irrespective of pre-existing glycemic status. Diabetes Res Clin Pract, 2022. 190: p. 109974., Singh, A.K. and R. Singh, Hyperglycemia without diabetes and new-onset diabetes are both associated with poorer outcomes in COVID-19. Diabetes Res Clin Pract, 2020. 167: p. 108382).

SARS-CoV-2 induced cytokine storm is a highly inflammatory and prothrombotic condition that can affect pancreatic β-cells both directly and indirectly (Khunti, K., et al., COVID-19, Hyperglycemia, and New-Onset Diabetes. Diabetes Care, 2021. 44(12): p. 2645-2655).

Infection of β-cells in COVID-19 patients results in reduced insulin levels and secretion, as well as cell apoptosis. Viral entry leads results in pancreatic endocrine dysfunction characterized by reduced insulin secretion in response to glucose and triggers the self-destruction mechanism of insulin-producing cells (through kinase pathways) (Khunti, K., et al., COVID-19, Hyperglycemia, and New-Onset Diabetes. Diabetes Care, 2021. 44(12): p. 2645-2655., Gerganova, A., Y. Assyov, and Z. Kamenov, Stress Hyperglycemia, Diabetes Mellitus and COVID-19 Infection: Risk Factors, Clinical Outcomes and Post-Discharge Implications. Front Clin Diabetes Healthc, 2022. 3: p. 826006). Thus, the cytokine storm in COVID-19 may exacerbate stress hyperglycemia as acute inflammation may further worsen insulin resistance (Khunti, K., et al., COVID-19, Hyperglycemia, and New-Onset Diabetes. Diabetes Care, 2021. 44(12): p. 2645-2655., Zahedi, M., et al., A Review of Hyperglycemia in COVID-19. Cureus, 2023. 15(4): p. e37487). Among these pathways, it has also been proposed that SARS-CoV-2 infection may initiate an autoimmune response targeting β-cells (Gerganova, A., Y. Assyov, and Z. Kamenov, Stress Hyperglycemia, Diabetes Mellitus and COVID-19 Infection: Risk Factors, Clinical Outcomes and Post-Discharge Implications. Front Clin Diabetes Healthc, 2022. 3: p. 826006., Tang, X., et al., SARS-CoV-2 infection induces beta cell transdifferentiation. Cell Metab, 2021. 33(8): p. 1577-1591.e7., Wu, C.T., et al., SARS-CoV-2 infects human pancreatic β cells and elicits β cell impairment. Cell Metab, 2021. 33(8): p. 1565-1576.e5). Last, microvascular dysfunction is a probable cause of hyperglycemia in COVID-19 patients as it may hinder glucose disposal. This relationship between microvascular function and glycemic control is reciprocal and well established (Stehouwer, C.D.A., Microvascular Dysfunction and Hyperglycemia: A Vicious Cycle With Widespread Consequences. Diabetes, 2018. 67(9): p. 1729-1741).

Reviewer 2 Report

The manuscript entitled “Glycemic dysregulation, inflammation and disease outcomes in 2 patients hospitalized with COVID-19: beyond diabetes and 3 obesity” by Angelos Liontos et al. showed that elevated CRP, IL-6, TRG/HDL-C ratio, and TyG index, severe pneumonia, and hyperglycemia in diabetic patients were associated with extended hospitalization of COVID-19. While lower levels of PFR within obesity patients were associated with longer hospitalization and a higher risk of death. These inflammatory markers and disease severity indices potentially explained why COVID-19 diabetic or obesity patients exerted higher mortality rates. The overall findings are interesting and meaningful. However, I suggest the authors showing the distributions of NLR, PCT, IL-6, PFR, TRG/HDL-C ratio, TyG index, or other important measurements with essential dot plots. Dot plots will display the differences of these markers more intuitively.

Minor editing of English language required.

Author Response

REVIEWER 2

Comment #1

The manuscript entitled “Glycemic dysregulation, inflammation and disease outcomes in 2 patients hospitalized with COVID-19: beyond diabetes and 3 obesity” by Angelos Liontos et al. showed that elevated CRP, IL-6, TRG/HDL-C ratio, and TyG index, severe pneumonia, and hyperglycemia in diabetic patients were associated with extended hospitalization of COVID-19. While lower levels of PFR within obesity patients were associated with longer hospitalization and a higher risk of death. These inflammatory markers and disease severity indices potentially explained why COVID-19 diabetic or obesity patients exerted higher mortality rates. The overall findings are interesting and meaningful. However, I suggest the authors showing the distributions of NLR, PCT, IL-6, PFR, TRG/HDL-C ratio, TyG index, or other important measurements with essential dot plots. Dot plots will display the differences of these markers more intuitively.

 Answer to Comment #1

We would really like to thank the reviewer for this comment. We have added to our manuscripts dot plots that display the differences of these markers per sub-group of patients (DM, obese, non-diabetic/non-obese). Dot plots are presented in figure 1, 2 and 3, respectively (pages 9, 10 and 11). In case figure quota is surpassed, figures 1-3 can be included as supplementary material.

Comment #2

Comments on the Quality of English Language. Minor editing of English language required.

Answer to Comment #2

We would really like to thank the reviewer for this comment. We have reviewed our manuscript and proper editing of English language was performed, where applicable.
